# Social Landscape, Peripheral Inclusion and Un-Practice: Concepts for Understanding Social Housing Daily Life in Open Spaces

**Veronica Garcia Donoso** [1,2,*] and **Eugenio Fernandes Queiroga** [3]

1   Architecture and Urbanism Course, Federal University of Santa Maria (UFSM),
    Cachoeira do Sul 96503-205, Brazil
2   Georg Forster Research Fellowship, Alexander von Humboldt Foundation, 53173 Bonn, Germany
3   Architecture and Urbanism Course, Faculty of Architecture and Urbanism, University of Sao Paulo (USP),
    São Paulo 05508-080, Brazil; queiroga@usp.br
*   Correspondence: veronica.donoso@ufsm.br

**Abstract:** This article presents new concepts for discussing urban social space, named "social landscape", "peripheral inclusion" and "un-practice". These concepts are based on the analysis of social practices in vulnerable neighborhoods with a high number of social housing blocks in South America. The aim of the article is to show that the complexity of social practices in vulnerable urban areas is not only the result of the urban environment, which combines social inequality, marginalization and insecurity, but also and above all of the management and maintenance of this inequality. The research method combines bibliographical research with the method of non-participant systematic observation, the latter analyzing everyday life in social housing areas of São Paulo-SP (Brazil) and Santiago (Chile). The discussion and results will lead the reader to understand not only the concepts, but also the idea that open spaces have an important role in social practices, especially public spaces. It seeks to demonstrate the importance of linking public spaces and housing in public policies for the creation of social housing, as opposed to housing policies that focus on the production of architecture disconnected from the urban and social reality.

**Keywords:** social landscape; open spaces; public spaces; social housing; social practices

## 1. Introduction

Urban inequality is a well discussed multidisciplinary study topic with a vast literature [1–3], many times associated with space analysis [4,5], since the physical distances and unequal distribution of opportunities in the urban tissue are often indicators of social misbalance.

The increased concern about inequality stimulates research around the globe. It is already clear that in the Global South, inequalities of income and social justice are often more pronounced than in the Global North. Even though the Global South faces high levels of poverty, the inequality is not only merely socio-economic-related, since there is a cycle of production and reproduction of inequality, many times associated with power and the threat to social contracts that underpin societies' democracy [6].

Researchers from the Global South discuss inequality from a point of view of the southern reality. With this considered, theories produced away from the local reality or based on the Global North's situation will have little explanatory power to many of the world's realities [6]. Henceforward, it is important to build conceptual thinking closer to the real world, especially considering the different ways of looking at and living in the Global South.

Inequality is also related with the concept of vulnerability, which emerged in the literature to take a step forward from the concept of poverty that had proved insufficient [7].

In this article, the vulnerability concept is considered, since it is a state of high exposure to risks and uncertainties, together with a reduced ability to fight those issues or to cope with their negative consequences [8–10].

Inequality is particularly related with the history of cities and their social groups, power disputes and individual thinking. Similarly, social practices in vulnerability contexts are also related with the historical, economic and social context [11]. They are also linked to culture and ideology. Thus, financial globalization has also generated more spatial and social polarization since the start of this century, sometimes related with migrations [12,13], economic crisis scenarios [3,4] and neoliberal city production models [14,15]. All of those elements are relevant challenges for Latin American countries [16,17]. Therefore, urban vulnerability is a complex issue that reveals the need to adapt public policies that can take a step forward to stimulate change.

Although vulnerability and urban inequality have been studied from many different perspectives, a zoomed-in analysis with daily life and social practices related to vulnerability is sometimes missing. Accordingly, studies from many countries have been developed considering the relationship between quality, comfort and the sustainability of social housing units [18–21], but not much is said about the open spaces in these areas.

In response to rapid urbanization since the 1960s, the production of social housing blocks has been common in many countries. Since then, many buildings have been built as part of mass housing projects. Often, open spaces were not designed and tended to be neglected in these housing projects, which were mostly designed as leftover spaces. A succession of buildings was often associated with a 'no neighbors' experience, as the streets were transformed by the gates and walls of the buildings. This was the case in many countries, such as Brazil [22,23], Chile [22,24], Mexico [25], Peru [26] and Ecuador [27].

Latin American countries have some peculiarities when it comes to social housing policies. In Brazil and Chile, since the military dictatorship periods, social housing policies have been developed with a view to mass production. In Chile, social housing has been developed with neoliberal programs since the 1970s. In Brazil, the neoliberal policy perspective began in 2009. For both, the challenge is to move from quantitative to qualitative housing production while responding to the growing housing deficit.

The challenges of social housing are not unique to Latin America. In Korea, for example, where more than half of the population lives in apartments, the highest buildings seem to have a higher rate of social pathologies, which could be linked to, among other things, the presence of fewer open spaces. A similar issue also appears in Japan [28,29]. In order to respond to the challenges of the growing population in Korea and Japan and also to the quality of housing, researchers are discussing other models of housing arrangements with the inclusion of small courtyards [28,30].

However, in terms of the interactional environment, courtyards or similar private open spaces do not represent a complete social practice, because the interaction is limited to the already known users of the space, without surprises that can embody complete social relations.

This article presents new concepts for discussing urban social space, called "social landscape", "peripheral inclusion" and "un-practice". These concepts are based on the analysis of neighborhoods with a high number of social housing buildings in South America. This was only possible through the observation of daily life in neighborhoods where social housing is concentrated in the Brazilian Metropolitan Region of São Paulo-SP and the Chilean city of Santiago, from 2013 to 2017 [22]. The choice of analysis for both countries was mainly due to the creation of the "Minha Casa Minha Vida" Brazilian housing program in 2009, inspired by Chilean neoliberal housing policies [23,24,31].

These policies accentuated the creation of social housing neighborhoods in areas of urban expansion, with very striking and similar characteristics, maintaining the construction of concentrated housing blocks in a condominium model as a way of making social housing. However, the question that arises is this: would it not be the time for this productive logic to be changed, turning it into more appropriate solutions for the daily lives of families and

cities? The answer to this question stays in questioning the condominium model for social housing and focusing on open spaces and public areas to stimulate social practices.

Housing policies, mainly influenced by mass production, have meant urban impacts and also impacts on the daily lives of families. Exclusion situations, lack of identification with the neighborhood and neighbors, fears and concerns, are issues that were observed in Chile and Brazil. Those issues coincide so that social practices happen mainly inside the home or in reduced and controlled living spaces, and not in collective and public spaces.

Chile has developed interesting neighborhood programs in an attempt to minimize the impact of the neoliberal production of social housing. Even if they are not yet able to change the overall picture of the inequality scenario, it is a process of paradigm change and a shift from massive housing production to habitat production [22]. However, despite the progress with neighborhood programs, Chile still follows the condominium model, called co-ownership, for the social housing architecture. The same is the case in Brazil.

The condominium model for social housing also increases this lack of identification with the living areas and the prioritization of individual social practices. This model appears as a solution for housing policies in both countries, being present mainly in metropolitan regions in the form of housing blocks.

It was found [22,32] that this model has a great impact on open space social practices, as the differences between the self and the other are accentuated. Therefore, social relations are limited by walls, rules and similar economical characteristics in the condominium models. This limitation, verified in both countries, led to the creation of the concept of un-practice, which represents an incomplete social practice.

This article is organized into three sections that explore the concepts of social landscape, peripheral inclusion and un-practice. These key topics are important to understand the social practices in vulnerable urban spaces.

The first concept, social landscape, states that the landscape should be analyzed through its interconnected social dimensions. The open space system has a great importance for citizens: it can be the space of forests and other green areas with a relevant contribution to the environment, but it is also the space of parks, squares, paths and streets. All together contribute to social relations.

Despite its relevance, open spaces are often disregarded in social housing policies, which mostly focus on architecture and sometimes on the urban context. This study aims to fill this gap in the literature concerning landscape concepts for the transformation of social housing spaces. It seeks to demonstrate the importance of considering open spaces in social housing policies.

In this regard, the second concept, peripheral inclusion, gives a new approach to the well-known concept of social exclusion [33,34]. The latter is related to the challenge for vulnerable social groups to overcome the context of vulnerability. However, exclusion is a word that indicates that those groups are not part of the economic cycle. As matter of fact, they are part of the global economy, nevertheless located in the peripheral area of it [22].

The third concept, un-practice, arises from the limitation of social practices: on the one hand, there is the transforming potential of open spaces through daily appropriation, which surpasses everyday life. On the other hand, however, there is the socioeconomic and cultural context, associated with subjectivity and individual conduct, which makes it difficult to overcome the alienated daily life. As a result, social practice is limited, incomplete, i.e., un-practiced.

The concepts of peripheral inclusion and un-practice are related. Both are equally situated within the larger concept of the social landscape. Spatially, social differences become visible in the urban space of cities through the logic and politics of land use. There is a paradoxical relationship between inclusion and the invisibility of difference. In other words, social differences coexist in urban space within the social and economic complexity of populations, but are distanced by society's fear and indifference towards socio-economic–spatial discrepancies. It is in this absence of differences that the un-practice occurs, with the closure of the complete social experience. In this sense, the concept of

un-practice is not without peripheral social inclusion, and both depend on an expanded view of the social landscape to be understood.

The original contribution of this article lies in this conceptual analysis, which has been specifically designed to address the realities of social relations in the vulnerable urban spaces of Latin America.

Although the concepts are based on the observation of Latin American realities, the results provide recommendations that may be suitable for the Global South and North discussion on how to create a better quality of life in cities with high vulnerability. The recommendations reveal the complexity of daily life in the context of vulnerable urban areas and provide guidelines for implementing neighborhood policies that integrate open space design with housing units.

## 2. Methods and the Theoretical Background

The method of these conceptual theories is bibliographical research and systematic observation with overt, covert and non-participant methodology [35]. This means that the study was developed with observation of the daily life of participants with the support of some members of the social group. It intended to avoid any kind of influence on the behavior of the local community, to capture the essence of daily life. To support the observation method, some questions were posed to the members of the social group that were aware of the researcher's presence.

The observation was carried out during PhD. research in two countries, Brazil and Chile. The research analyzed social housing buildings in neighborhoods where social housing was concentrated in the Metropolitan Region of São Paulo and in Santiago, from 2013 to 2017. In Santiago, case studies were conducted in the municipalities of Bajos de Mena, Santa Adriana and Cerro Navia. In Brazil, some housing condominiums of the "Minha Casa Minha Vida" program were analyzed in the Cidade Tiradentes neighborhood and also in Guarulhos, a city in the São Paulo metropolitan area [22].

For the case studies, the spatial syntax method was used to understand the spatial configuration, urban morphology and their relation to the social practices of the people [36,37]. This method was used to study the exterior spaces of the social housing blocks especially, as well as the neighborhood streets. Spatial syntax studies have shown that places that are more accessible in spatial configuration are generally and potentially the spaces where more interaction between people takes place [38,39].

Considering the theoretical background of the analysis, some authors in the fields of psychology and psychoanalysis were key to this work: Carolina Besoain [40], a Chilean psychologist, who reflects on the subjectivity that exists in the processes of acquiring a house; and Christian Dunker [41], a Brazilian psychoanalyst, who addresses the psychoanalytic aspect of condominium life. Both deal with strangeness in relation to the other, which explains the subjectivity existing in social practices in open spaces.

The theoretical background also crosses the field of sociology, based on the work of Bourdieu [42,43], who places social genesis in the structure of relationships, guided by mechanisms of domination. Although in another context, Bourdieu's construction is also anchored in subjectivity, since individual behaviors in his analysis are dependent on a system of relationships. His concept of *habitus* deals with the domination and struggle of social groups that unconsciously follow certain individual and collective behaviors.

The Latin term *habitus* originates from the Aristotelian notion of *hexis*, which, in turn, designates the set of dispositions of agents in which practices become the principle that generates new practices [44]. These dispositions are organized in systems and regulated by a socially structured ambience, that structure and are structured, being the principle and product of the agents' practices, perceptions and actions, a "habitual state" [43]. This structured and structuring ambience can be both the physical and affective surroundings of social and class relations, which guide and determine social practices.

For Bourdieu [43], social life can be analyzed both from the set of geographical conditions that define unequal individualization and the combination of social groups in specific

fields, as well as through the analysis of territorial fields where different processes of local identification occur. In this way, social life happens spatially according to the productive forces of the relations of production and reproduction of agents. Daily life implies social relationships and networks. Daily social analysis is an exchange of knowledge between the observer and the observed, used to understand the practiced territory from all the agents who practice it.

Goffman [45] analyzes everyday life from a theatrical view, in which all subjects are actors playing a role, or even a character, defined by the system and by the environment in which they find themselves and with which they interact. Although related to the Anglo-American society of the 1970s and with a framework focused on internal ambiences, many of Goffman's analyses can be applied to current external spaces, as they relate to the social rules defined and supported by the users themselves.

Thus, for Goffman [45], the expressive component of social life is given by the impressions presented and received by the other, in a created game of representations that, in most cases, are falsely represented.

This theatricality of everyday life is related to social reproduction, a process that is guaranteed by relations of domination with the unconscious of maintaining social structures, although with remote possibilities of escaping that same social structure.

This reproduction guides practices and legitimizes conditions of domination, in a cyclical process without changing the pre-established condition. This process, for Bourdieu [42,43], is related to the *habitus* and the *field structure* that, although holding subjects in a preconceived reality, does not prevent the overcoming of their condition.

The *habitus*, as a structured structure, is a social product, acquired in an innate way. As a structuring structure, it is a social producer, which operates unconsciously, guiding practices and the social space.

As product and producer of the same story, incorporated as a system of dispositions, *habitus* works in relation to the social field and is part of the individual's conduct that translates itself in attitudes, manners (*hexis*) and moral appraisals (*ethos*), retained and reproduced in social practices.

There is a social *habitus* [42,43] that guides the way of acting and thinking about everyday life. This conduct is governed by a hegemonic cultural code that is often deeply hierarchical and elitist. As a result, the critical and political awareness of architects and urban planners is often unconsciously governed by an elitist codification, according to which it is imagined that a part of the population wishes to possess what other social groups have.

It is the same relationship between yourself and the "other", when the "other" is only understood through a look at oneself. The greatest difficulty is to disconnect from this hegemonic universe that permeates the dialectic of the self and the "other", allowing ideas for landscape and urban planning to be carried out from new proposals and free from the bonds of the dominant cultural codification [46,47].

Specific social fields are analyzed as areas of socialization, in which agents, consciously or unconsciously, relate politically, economically, socially and symbolically from themes of specific interest.

In a Bourdieu metaphor, the structure of the field is like that of a game, where each player has cards from different types of capital—economic, cultural, social and symbolic—which defines their strategic position [43]. Thus, social fields are defined based on areas of interest and class positions and are determined by historical factors in the formation of social life, being, therefore, contradictory, and marks of privilege of specific social agents.

Hence, the relationship of domination (and the reproduction of domination) between agents defines the social structure, considered as areas of socialization, in which agents, consciously or unconsciously, are related politically, economically, socially and symbolically from themes of specific interest.

The *habitus* directs unconscious actions, originating from a position occupied within that social structure and from a conditioning that stimulates the reproduction of social circumstances, guiding actions from an unconscious internalization of social position, status, class, gender and ethnic origin, among other aspects. In this sense, all daily actions are socially situated. The *habitus* manifests itself in social relations, actions, sports practices, way of dressing and speaking and posture, among others.

In relation to social analysis, the work of the Brazilian sociologist Ana Clara Torres Ribeiro [46,47] was also important. An interlocutor of Milton Santos, a well-known Brazilian researcher [48–51], she was the main reference for the reflection of daily practices in this research. Her work left a huge legacy on the construction of space, class struggles, conflicts, collective consciousness and the marks of processes in a territory and society.

It is from these contributions of sociology, psychology and landscape that this article's logic about social practices and daily life in social housing was developed.

## 3. Social Landscape

Landscape can be analyzed from several social dimensions that interpenetrate, as it is an expression of society, revealing its customs and social characteristics. Thus, landscape is the dynamic result of the interaction between social processes—economic, cultural and political—and natural processes, that are in constant change [52,53].

Additionally, it is simultaneously a result and part of the processes that occur in it, expressing the desires and limitations of a society in modifying a territory for its activities.

From the point of view of social practices, landscape planning for open space qualification has a great importance.

Social practices can be directly associated with the availability and quality of open spaces, and it is in the unconscious everyday life [54,55] that action and discourse come together in the moment of human coexistence. Action and discourse are only manifested if there is a Public Sphere [56], in the sharing of actions between human beings together. Furthermore, it is together that the power of a social group is exacerbated against pre-established narratives.

The everyday Sphere [56] is the closest to social life, and it is there that the possibility of disruption of alienated production and reproduction is outlined. The association of daily life with social life places it in a relationship with the totality of the social structure, which allows its perception as a "hidden face" [47] of social life compared to the domination structure.

Open spaces [52] are all spaces without a roof or construction, regardless of if it is public or private space or if it has vegetation. Therefore, squares, parks, streets, backyards, vacant lots, forests, beaches and even agricultural areas [57] are open spaces of relevance to cities.

All open spaces are interconnected in a complex system, which interrelate with other urban systems complementarily. It is not the physical and visual connection that is most relevant to the system, as these spaces have multiple roles, both in environmental and social issues.

The Open Spaces System allows public interaction, the free movement of living beings and social interactions. This system may be due to specific planning or it may arise through spontaneous appropriation.

Those spaces represent the largest percentage of area in cities and, regardless of the environmental/urban quality and aesthetics, they have the potential to welcome the Public Sphere, be the state for political protests, popular festivals and others, from the daily encounter with the "other" [47]. Precisely for this reason, open spaces should be the focus of social housing policies.

This does not mean that they are not considered at the present moment in housing policies, but the emphasis has not been on generating open spaces for the daily practices of social groups, especially in the Brazilian case.

According to Segovia [58], some common characteristics can be identified in social housing blocks, which can be observed in several Latin American contexts: peripheral

location when built; disconnection between the urban design of the complex and the immediate surroundings; and higher density compared to the rest of the city. Considering these characteristics, it is assumed that residents are also disconnected, isolated from the city itself.

The maintenance of social housing production with these characteristics also shows a disconnection between academic analyses and political and institutional actions, resulting in the absence of strategies to effectively apply them. Building habitats with articulated open spaces, public spaces and urban infrastructure and equipment, creating neighborhoods and urban continuities, is not yet a common practice in social housing projects in Latin America.

## 4. Peripheral Inclusion

The history of many Latin American countries' urbanization is directly related to the history of socio-spatial segregation. The urban privilege of the middle- and upper-income groups in Latin American cities is fraught with historical heritage, which arranges private interests above the collective.

The differences between groups in the population are not limited to economic and social inequalities and access to better material living conditions. They are also the result of different levels of social resistance acquired from the history of inherited struggle. Thus, inhabitants of the same place do not always experience identical temporality, as they are unaware of these historical differences that are reproduced in social practice. This lack of knowledge, or even forgetfulness and denial, amplifies the risk of peripheral inclusion and resignation to social–spatial segregation.

Peripheral inclusion is a concept that aims to overcome the idea of social exclusion. The "exclusion" term brings the false notion that the "other" has no power with his own action [47]. In fact, what occurs is peripheral inclusion, since the "excluded" are, in reality, part of the system, in a bitter paradox of maintaining inequalities.

The wide use of the term "social exclusion" only seems to lament the social problems of socio-spatial differentiation without, in general, explaining the origin of the problem.

Social groups said to be "excluded" from the system are actually included, contributing to the maintenance of low wages and high levels of surplus value extraction in our capitalist socio-spatial formation. These social groups disadvantaged by the system are not only a reserve army of labor, but also a product of the inequality in the distribution of wealth generated, that is, they are part of the economic and social system and, therefore, cannot be simply called "excluded".

Thus, given the imprecision that the exclusion expression brings, many experts have discussed new ideas and concepts. Ribeiro [47], influenced by Milton Santos' concepts of "marginal inclusion" and "slow man", comments on the problem of the term "excluded": for her, this word implies that the "other" does not have power or critical awareness about overcoming the reality, which is not true. Maricato [59] states that the paradox between the underprivileged in the system, the patrimonial heritage, urban legislation and the power of classes or institutions, are parts of a "systematic contravention", which is the basis of the Brazilian urbanization pattern, according to which urban expansion and the expulsion of the population with fewer resources resulted in the growth of a hidden city. This invisible city is the one that is not seen and does not want to be seen, and includes, among others, families that live illegally. According to Martins [60], "excluded" and "exclusion" are terms used by society's beneficiaries, that is, those who are not included in the periphery.

Urban expansion controlled by political and economic interests made the distortion between real demand and the interests that govern urban processes even more striking [61,62]. Thus, the reproduction of territories of peripheral inclusion is part of a cycle that produces and deepens economic and social differences, as well as situations of vulnerability. Exclusion, therefore, is not a circumstantial accident.

The contradictions of everyday urban life constantly create situations of favoring and disfavoring, which are reactions of the system itself, with social groups participating in this

contradictory cycle caused by social, political and economic processes. In this sense, using the notion of "excluded" is incorrect, as the less favored social groups are not outside the system, but peripherally included, in a perverse process that maintains inequality.

## 5. The Need to Overcome the Condominium Model for Social Housing and the Concept of Un-Practice

Since the Brazilian and Chilean military dictatorships, a *modus operandi* of producing social housing blocks using the condominium model has emerged. The idea was based on a simplification of the guidelines of the Athens Letter for modern architecture principles. Even if the simplification of this model was highly criticized by experts [22,31], it is still the pattern used for social housing in the Latin American context [22].

The condominium model for social housing has been used mainly in metropolitan regions in Latin America, due to the high cost of the urban soil and the necessity to optimize the architecture to answer the high demand of the housing deficit.

The housing condominium is a realization of a housing ideal. This urban model produces a subjective security effect, which occurs because of the protective walls and also the coexistence between equals, that is, by social groups with similar economic power. The social housing condominium is a transposition of the middle- and upper-class condominiums. Therefore, to explain the social housing condominiums, it is important to first address the middle- and upper-class model.

In Latin America, the condominium model refers to a gated community complex [63]. In the Americas, gated communities have a long history, sometimes designed as walled or non-walled areas, but always associated with indicators of segregation. Real estate development and land markets use the condominium model as a product for exclusionary and often all-inclusive developments, anchored in the precariousness of cities and integrating homogeneous categories of space users/residents.

In cities with high levels of socio-spatial inequalities, the marketing of the condominium model coincides with levels of violence, and the population looks for gates, fences and other security measures to try to live a life away from reality. This urban model is not exclusive to the larger cities: as a real estate product, gated communities, condominiums and other walled market products are spreading to areas where there are no levels of violence to "justify" living inside fences.

In Brazil, walled-off enclaves are associated with high levels of urban violence, but especially with the fear of their inhabitants [64].

This urban model is growing in many parts of the world facing inequalities, particularly in the Global South. In South Africa, for example, there is a growing number of gated communities, many of which are on an urban scale and combine the concept of 'live, work, play' [65]. This has been also a well-known model for Brazil, as exemplified by the Alphaville concept [66].

The Brazilian condominium or the Chilean co-property urban model can be explained as a combination of the private and collective ownership of spaces. It is a system in which individual homeowners share the ownership, management and use of common areas. A condominium association is often created to manage the common areas, where rules and regulations for the use of the spaces, as well as some aesthetic principles for the houses and gardens, are discussed among the residents. Whether the condominium is for high-, medium- or low-income individuals, the result is class segregation, with exclusionary mechanisms preventing other social groups from entering the gated complex.

The exclusionary mechanisms are those created to make the social difference invisible to higher income groups. For example, in countries such as Chile, Brazil, Argentina and Peru, high-income housing real estate products have several strategies. Separate entrances to the gated complex, and even to the house and apartment, are common, to avoid encounters between the rich homeowners and low-income workers such as housekeepers and security guards. Separate lifts are also common. This condominium model can be vertically oriented as a building or horizontally configured as a gated community of houses.

Although usually associated with higher income groups, the condominium as a gated community is also common in social housing projects.

There is another complex situation related to the condominium model in Brazil: Not every condominium is a legal condominium, that means, approved by legislation. It is common to have regular urban settlements of residential zones that are later fenced in by the residents themselves. This is usually conducted with the approval of the local council. This means that gated streets appear in a regular neighborhood, creating barrios and residential enclaves [57].

Condominium life, regardless of the social group, tends to exclude what is outside the walls of everyday social practices. The condominium concentrates the coexistence in the intramural space, in a space conceived and lived as a false protected universe that shelters a fictitious freedom.

Gated communities, especially high-income ones, are associated by the dominant media with an image of happiness, ironically in an illusion of urban reality. The artificialization of high-income gated communities, empty of urban content, is derived from planned and controlled forms of living in utopian cities, with regulations, protocols and the promise of recreating an idealized experience or way of life.

This artificial urban reality, based on the prototype of a high-income condominium, is also a morphological model for social housing, stimulated by several reasons, including:

- the reduction of public costs by directing the creation of living spaces within the housing estate to the private sector;
- the false belief of managers, financing banks and other technicians that this is the best morphological model for collective living;
- the "modus operandi" of production, following a practice that has been established for years, of reproducing construction models, with low construction costs and minimal creative thinking.

In the condominiums of social housing, the impersonal authority of drug trafficking or other practices of organized crime is often added to the control of the walls that guide actions and challenge the orders of the condominium, personifying an authority constituted by the imposition of fear that demands submission and deals with exceptions to the rules, both intramural and extramural urban, as will be seen in the case studies in Section 6.

The fact is that social housing in the form of condominiums has become established as a housing policy, increasingly seeking, in walled enclosures and reduced living, to manage intractable socio-spatial differences and to mitigate the lack of urbanity of the places in which they are located.

The condominium model in social housing is complex and requires a lot of social work for the families to adapt to the model of sharing spaces and common costs, as the case studies in Section 6 will show. This is even more critical in large-scale social housing condominiums, with several buildings in the same condominium area, but also in single-family housing units in the condominium model, which are stamped ad nauseam in areas of urban expansion for low-, middle- and high-income social groups.

It is a mistake to impose urban models from the middle and upper classes on the lower classes, uncritically accepting the behavior of dominant classes as a successful manifestation of civilization and urbanity, without investing in the definition of spatial structures that value the urban experience of the "other", and without reproducing either models of privileged areas or of poor areas.

Dunker [41,67] highlights that it is necessary to consider the city as a principle, where territories are not excessively defensive or individualized, with ownership relations divided by social class or income.

The relationship with the other needs to be established: the recognition of differences allows what Dunker [67] calls "productive experiences of indetermination", in which the indeterminate, the unknown, leads to the possibility of overcoming established patterns. The undetermined is the subject that is not interpreted by its appearance, speech, body

movements or any social group or class, eliminating the prejudice of the social experience. For the psychoanalyst, indeterminacy is one of the principles of the city.

However, as a defense against the undetermined, in the name of the insecurity that the strangeness of the other causes, protected spaces are created, concentrating groups with similar interests, where the conflict with the undetermined can be managed. The psychoanalyst brings a set of malaises that characterize contemporary society and which are part of the psychopathies of urban life: insecurity, intolerance and isolation from others [41].

To the author, the condominium is created to prevent those feelings, as an association of contemporary malaises avoided under management. What defines the condominium is the administration of usage and actions in the space, following rules and principles that seek to avoid a complete and indeterminate social experience. Thus, the model can be for housing, business, commerce and even health, like hospitals. With determined actions and based on fears and rules, the condominium classifies and isolates social groups, excluding others.

In condominium models, private open spaces are planned for common use by residents of the same condominium, where collective practices are restricted by walls. This occurs independently of the income of the housing condominium. Outside, there are no greater interactions between groups, and the relationships of fear of the other are reproduced.

The un-practice represents the dialectic that such reflections have stimulated: on the one hand, we have the territory with its possibility of transforming itself through use, belonging and combating alienated daily life; on the other, we have the neighborhoods where social housing is concentrated, a difficulty in bringing people closer to public spaces and social practices.

Thus, the un-practice refers to a limited practice. It is not that it does not exist, but it occurs incompletely, due to the vulnerability and subjectivity present in the fragile social context, and also due to the imposition of urban models, norms of conduct or control by leaders within the social group.

Some values need to the considered for the production of urban spaces in the context of vulnerability, among them are the rescue of history and the singularity of Latin American social production; the interdisciplinary understanding of the urban issue; the inclusion of the "other" through participatory processes; and the resistance to an uncritical reproduction of paradigms and models.

Little or no social participation in the design of public policies and in the definition of the needs of social groups; control by few community leaders; the imposition of urban models; urban insertion in contexts of social, economic and cultural vulnerability; a weak relation between the housing internal space and the public or collective spaces; and between other dilemmas, make social practices limited and incomplete, that is, un-practiced.

The complexity of daily life in social housing with different types of suffering, such as isolation; peripheral inclusion; loneliness; concentrated power by groups and leaders; and the permanent feeling of strangeness, insecurity and violence, are related to the social housing massive model production. The growth of urban violence, hostility between groups and fear of using public spaces is not only result of the social inequality, but especially of its maintenance [41], which is a public responsibility topic.

## 6. Empirical Analysis and Its Relation to the Concepts

Although this paper does not aim to explore the case studies due to their complexity, the concepts here proposed can be better understood through some examples.

As already presented in the discussion, in many countries, social housing policies are characterized by a way of producing housing on a large-scale and using the condominium model. This characteristic was a modern utopian solution to face the increasing housing deficit, but it represents the sheer ineffectiveness of imposing a design model for a social group.

Morphologically and architecturally, the blocks in each country have similar characteristics, such as the size and shape of the building, the proportion and layout of the



apartments and the common areas in the condominium model. Usually, the selected families are registered on social programs. Depending on some political and local peculiarities, the families receive housing according to the availability of new constructions individually or sometimes as a group. The location of the housing depends on several factors, sometimes related to government regulations and sometimes related to the availability of the real estate of companies. In both cases, the location of the product is related to the real estate and urban land market, the second case presenting more disadvantages for social housing due to the cost of the best located urban land.

Two case studies are presented to illustrate some of the particularities: The Brazilian housing block "Parque Estela Residential Condominium" and its neighbor "Vila Pimentas Residential Condominium", in Guarulhos-SP [22]. Both were created with the "Minha Casa Minha Vida" program, but in different modalities, the first as "Entities" (PMCMV-E) and the second as "Enterprises" (PMCMV) (Figure 1). The "Entities" modality was formatted in a similar way to the "Enterprises" modality, with the difference being that non-profit organizations, duly qualified by the Ministry of Cities, could be in charge of supporting the social group in the housing process.

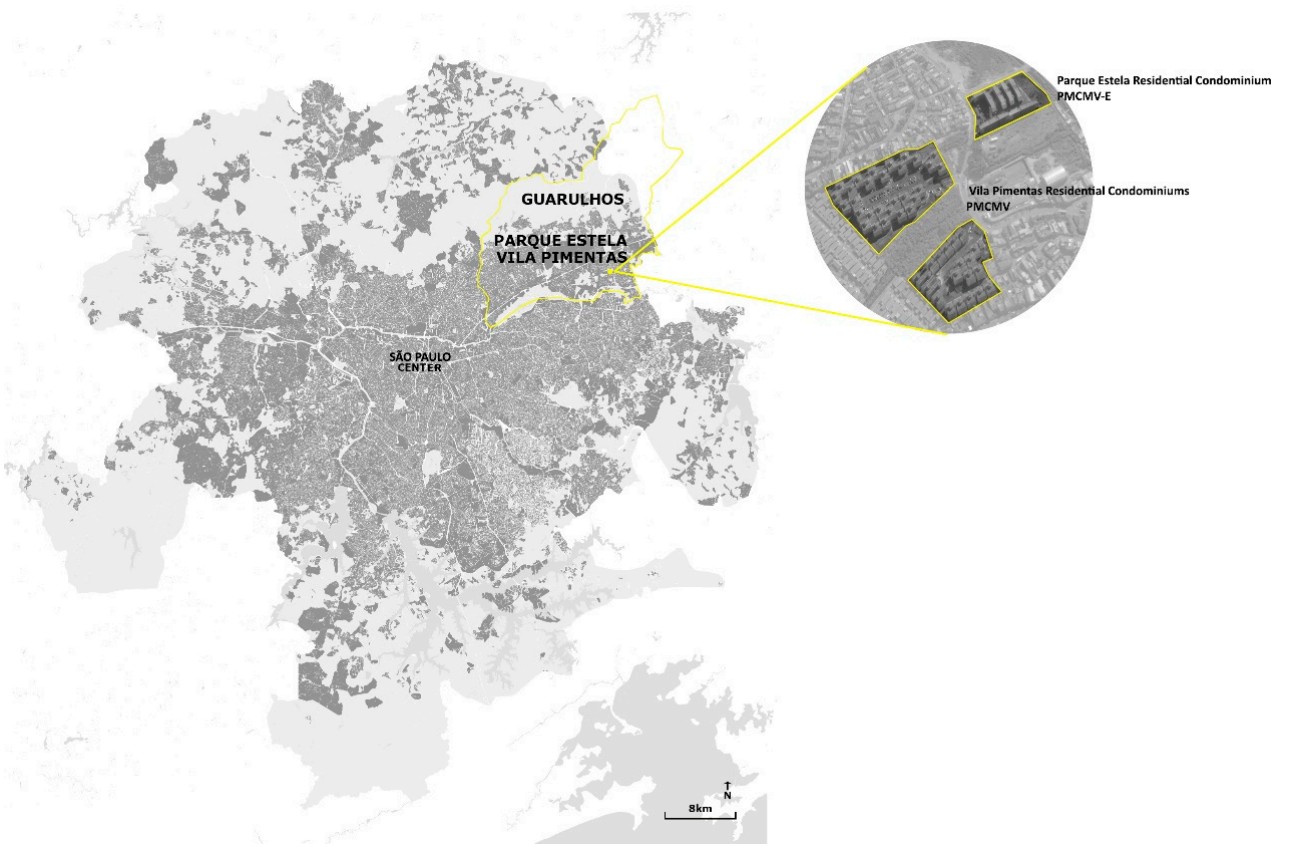

**Figure 1.** Location of the case studies in Guarulhos-SP, Brazil.

The case studies reveal the limitation of social practice in the intramural everyday life of social housing complexes, structured by the morphology of the condominium buildings, which directs practice to areas within the company, reducing daily coexistence—and conflict—to the neighbors themselves.

The un-practiced territory, whose social practices are not fully realized due to various limitations, is the result of the historical context of the production of urban space, with a direct relationship to social, cultural, economic and ideological issues that characterize a certain social order.

Physical and subjective situations guide social practices, conscious or not, in the open space. The lack of use and the strangeness of spaces for collective living in social housing

are aspects of the limitation of un-practice, where individual behavior prevails to the detriment of collective behavior, whose conflicting processes are marked in the territory and in society.

"Parque Estela" was conceived together with the National Movement for the Fight for Housing ("Movimento Nacional de Luta pela Moradia"), a very active movement in the metropolitan region of São Paulo, and with the technical consulting NGO "Peabiru", which advised the social group from the bureaucratic aspect of the program entry to the definition of the architectural project.

"Parque Estela" consists of six residential buildings within the same condominium, five of which have five floors and one has six floors, for a total of 218 apartments of 41.35 m$^2$.

The architectural project was developed by the Peabiru team without the participation of the social group, since it depends on the regulations of the housing program, which do not allow much flexibility in the definition of the architectural model. On the other hand, participation was possible in the definition of the open spaces of the condominium, characterized by the open-air parking area and other open spaces for community living: a barbecue area, benches and tables, a playground and a community center (Figure 2).

**Vila Pimentas Residential Condominium**

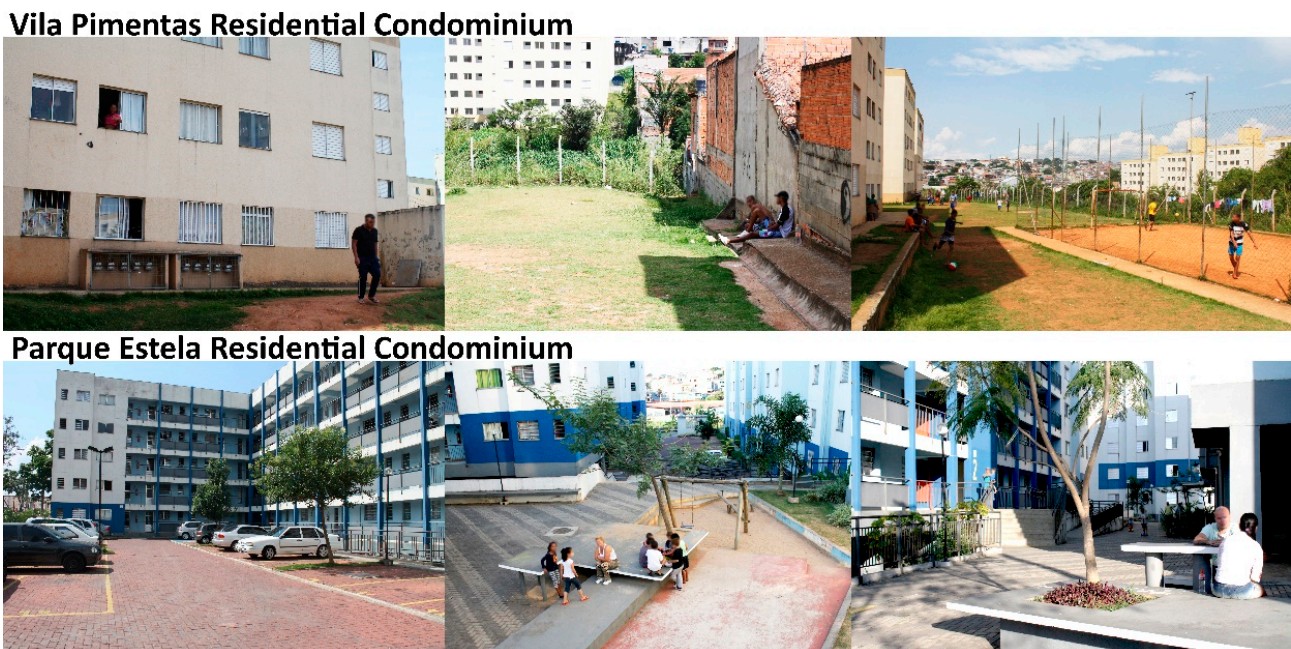

**Parque Estela Residential Condominium**

**Figure 2.** Open space areas in both case studies. Author's photographs.

Most of the residents had never lived in a condominium before. A community leader spoke with the author about some of the conflicts that have arisen: disputes between neighbors, looting and resistance to established norms.

There is also a dilemma in working with condominiums in social housing, which is the cost of maintenance. Gardens, for example, have a cost.

There is an illusion that moving to an apartment is the solution to many anxieties, due to a lack of knowledge about the routine costs and the limitations of the condominium model. In general, families believe that by distributing all the costs among all the neighbors, it is possible to have spaces like those that are mediated for higher-income classes. However, the management of condominiums in social housing involves a complexity typical for the social group, since there are still significant differences in income within the families.

On the one hand, in the design of collective spaces, there is the possibility of building social bonds, even if they are limited to the social group and an un-practiced territory. On the other hand, it is in collective spaces that the main problems of coexistence arise, the solution to which requires daily exercise in recognizing a space that belongs to everyone and that welcomes differences.



On the other side of the street, there are the "Vila Pimentas Residential Condominiums", which are divided into two different condominiums that were both built with the "Enterprise" version of the housing program. The families that received housing in Vila Pimentas lived in different favelas of Guarulhos. For example, Vila Pimentas I, with 580 housing units, brought together residents from three different favelas, causing an immediate conflict created by the combination of social groups with different histories and characteristics.

As example of the history of the families, there is the history of the formal population of the "Vila Flora favela", known as "Itapegica", a group of almost 500 people. They suffered from several problems, such as fires and contamination of the soil where the favela was located, and needed to be relocated.

In an interview with the author, a community leader said that women were more involved in moving from the favela to the Vila Pimenta condominium because the community environment was very unhealthy, with pollution from a lack of basic sanitation, insect and rodent infestations, and floods and fires. The women no longer wanted this kind of environment for their children's education.

The community leader said that the move to "Vila Pimentas" was initially positive, as it fulfilled an individual's dream of owning a home. However, after a short time, problems began to arise due to the mixing of different social groups and the social problems that already existed in the favela, which were exacerbated by the rules of the condominium model: loud music and "baile funk" behavior, drug trafficking, conflicts and fear among the residents.

In the beginning of the establishment of the condominium, which included the distribution of costs, there were already many conflicts between the members of the community, with situations of domestic violence and family disputes in the open space areas.

Despite the difficulties in the favela, the community leader notes that there are still many residents who would have preferred to stay there rather than move to the condominium. This preference of some members of the social group occurs because in the favela, there were no extra bills to pay and, above all, no rules. As a result, many residents have not adapted to life in the condominium and are nostalgic for life in the favela.

"Vila Pimentas I" has 29 housing blocks, each with a community manager, in addition to the building manager and the condominium sub-manager. In the common areas of the condominium, in addition to the areas reserved for parking, there are several remaining and unqualified open spaces (Figure 2). A wooden swing and a sports field are the only social elements built in the open spaces, but some appropriations can be seen, such as small gardens made of painted tires and plant species planted by the residents themselves, close to their blocks.

Conflicts are aggravated by the presence of drug trafficking within the condominium. One of the most remote blocks, in a privileged position to control the entire open space of the condominium, is under the control of drug traffickers, which unfortunately limited the field research.

As shown in the example, the imposition of the condominium model does not bring good results for social practice, which is aggravated by moving to a place with unknown neighbors, unplanned monthly expenses, and collective spaces that do not represent the needs of the residents.

As for the common areas in condominiums, there are few ways to adapt them to the daily lives of families. Exercise equipment, barbecues, hammocks and kiosks, inspired by landscaping of higher-income condominiums, are a misguided transposition of the middle- and upper-class housing model to social housing. Initially, these areas are viewed positively by families because of the desire for spaces seen in the promotion of higher-class developments. Later, however, the cost and difficulty of maintenance, as well as the lack of identification with the spaces, cause them to deteriorate, leaving only empty spaces.

In general, there is a fear of using the space outside the walls, a situation felt by the researcher herself, who was always invited by residents to park her private vehicle

inside the condominium for security reasons. The researcher was also always accompanied by residents during field visits, even if it was only to cross the street to get to the other condominium. Residents fear some clashes with drug leaders who are positioned in strategic locations both between the condominium walls and within the condominium areas (Figure 3).

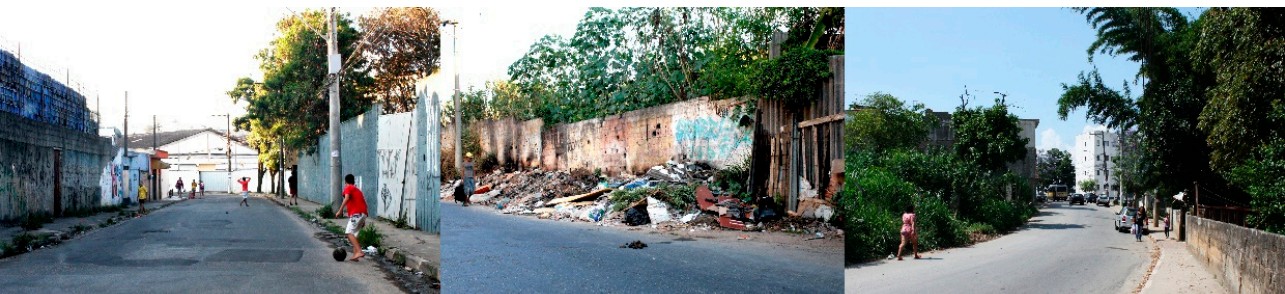

**Figure 3.** The streets surrounding the condominiums "Parque Estela" and "Vila Pimentas". Author's photographs.

This relationship of fear and control is more complex in social housing condominiums than in others with different social classes. In social housing, it is common for groups from different cultural backgrounds and origins to move together to the same condominium. This also brings together different leaderships of social groups, and the structure of relationships can be oriented towards mechanisms of domination that structure the *habitus* [42].

The condominium is a kind of an autonomous urban unit, which replaces the lack of urbanity in social housing surroundings. However, the social housing condominium is governed by rules and exceptions to them, which lead to losses of a complete social experience due to the isolation of individuals and the un-practice.

In condominiums, differences are resolved through a subtle code of coexistence, in an invention of a common life without a real community, that is, an un-practiced, invented, controlled and idealized everyday life limited to the same social group. The condominium represents a new policy for the management of differences, conflicts and social antagonisms, which requires a leader figure who proposes the ideal of coexistence in order to manage conflicts and make everyday life possible. The leader is sometimes a community leader, sometimes the organized crime leaders, who perform control and have a power status.

The social housing condominium represents an "included" poverty, that reproduces and maintains itself in the same place, but in a controlled space. Concentrating families with the same economic profile also brings together social issues. Neighborhoods where social housing is concentrated have accentuated social problems and fear in the use of open spaces, in addition to encouraging the persistence of peripheral inclusion.

## 7. Conclusions: The Importance of Considering Open Spaces in Social Housing Policies and Overcoming Peripheral Inclusion and Un-Practice

As presented in this paper, the peripheral inclusion and un-practice concepts are both interconnected and related to the inequality and vulnerability in cities. The peripheral inclusion of the other in social practice causes the un-practice. The concept can be extended to different situations when the fear, exclusivity and invisibility of social groups are presented.

In the case of Latin American social housing, this limitation is related to, but not limited to, the use of the condominium model. In some countries, it is possible that the condominium model will be adapted to social housing and there is no need to change this reality. However, in Brazil and Chile, as in some other Latin American countries, such as Mexico and Peru, the condominium as an urban solution is a model to be discontinued, especially for social housing.

With regard to the condominium model or other urban solutions that are transferred from high-income to social-income groups, it is important to emphasize that the imposi-

tion of urban models for social housing is a mistake, aggravated by the urban insertion in contexts of social and economic vulnerability and also by management without the participation of the population in the defense of their interests.

Although appropriation is not always related to the quality of spaces, there is a need to establish qualitative criteria for their analysis. However, it is complex to define them without falling into the pitfall of imposing an external solution, which can be developed in an imposing way as a mechanism of social control.

However, the new housing areas that remain marginalized will be doomed to failure, not only because of their location, but also because they are on the margins of the elaboration and construction process. Thinking and producing social housing from an external perspective, without understanding the specifics needs of families, attacks the freedom and individuality of the social group.

Current social practices in social housing are highly influenced by subjectivity and individualism, since the stimulus caused by the individual struggle for access to housing has particularized the celebration of collective achievement. The achievement of each family is celebrated individually, and not the conquest of the social group [32], except for the housing produced by organized families.

Accordingly, the ways to overcome peripheral inclusion and un-practice are not unique and depend on different situations. Here are four guidelines to help overcome them: a fair city, neighborhood creation, rights to city and landscape, and political consciousness.

The first guideline is to create a fair city, that is, a city in which identity and dignity are respected; equal opportunities are guaranteed; popular participation in the creation of social norms and urban policies is assured; the use of public spaces for democratic practices with free expression is recognized; differences coexist without discrimination, marginalization and stigmatization; the social economy is promoted with public resources and the equitable distribution of wealth; the use value of urban space is recognized above its market value; and the spontaneous and free expression of people is allowed in their urban experience [22,46].

The second guideline is to create neighborhoods with a mix of functions, construction typologies, as well as variations of open spaces, public infrastructure and the necessary facilities for social group activities. The focus on the neighborhood will allow the passage from "habitat" to a "lived space". This may sound very basic for countries in the Global North, but for the Global South, it is still on the to-do list. A culture of taking care of common areas also needs to be implemented.

Considering the creation of a variation of open spaces, this is an urban design solution and strategy to meet both social and environmental needs. Open spaces should be planned as a system, where spaces relate to each other to provide opportunities for human activity and movement. Public squares, sidewalks and other public open spaces should be provided in addition to the private open spaces of courtyards, with a variety and combination of open spaces to encourage social encounters and street life. On the other hand, the practical success of the spatial design depends on many variations, and the cultural habits of the inhabitants should be taken into account, as well as the bioclimatic characteristics.

Often, the continuity production of housing neighborhoods without articulated open spaces does not allow a scenario that stimulates an effective relationship with the place. The importance of open spaces in urban design is precisely the possibility of articulating constructed elements, which include housing units, building condominiums, commercial or mixed-use areas and urban equipment, among others, through the creation of squares, accesses, sidewalks and vegetation. These elements can establish a visual unit for the space users, which facilitates identification with the surroundings and encourages their use and appropriation.

In general, what is observed is that open spaces are treated as remnants in areas with social housing, which are not relevant to generating a positive spatial identity.

The scale relationship of the blocks is also essential. It is possible to design a housing neighborhood that allows spatial identity for the users. However, in order to do this,

a social function of conviviality, circulation and appropriation of open spaces must be considered, and demands must be made for them to be designed and built.

Open spaces are directly associated with the territorial and spatial context, which affects the daily lives of users. Therefore, they should be parameterized following requirements that qualify urban design, focusing on their social function, to achieve the creation of a habitat.

This is also related to the third guideline: rights. The right to the city [68] is also the right to the landscape [69].

Public spaces have the ability to strengthen the public sphere. Because of this importance, they must be taken into account at many political levels. The relevance and appreciation of the public dimension of space includes the search and legal action for rights beyond the right to the city, including the right to landscape, places, environment and territory [69].

The fourth and final guideline is the need for political awareness to implement socioeconomic changes. This includes dedicating effort and investment to the design and maintenance of public open spaces. This should be considered from the perspective of what is best for the social group and for the future of mankind. Many categories should be taken into account: citizen participation and social networks, the management of spaces, urban insertion, microclimate adaptation, morphological diversity and functional flexibility, visibility and dimension [22].

The four guidelines are connected. Addressing them could have a positive impact on the quality of life and the reduction of urban problems caused by inequality in cities.

**Author Contributions:** Conceptualization, V.G.D.; analysis, V.G.D.; writing—original draft preparation, V.G.D.; writing—review and editing, V.G.D.; supervision, E.F.Q.; funding acquisition, V.G.D. All authors have read and agreed to the published version of the manuscript.

**Funding:** This research had the support of The São Paulo Research Foundation (FAPESP, Grant numbers 2013/04592-0 and 2015/07233-6) and the Alexander von Humboldt Foundation (Georg Forster Research Fellowship for Experienced Researchers).

**Institutional Review Board Statement:** Ethical review and approval were waived for this study since the method was based on systematic observation. Participants were not in risk during research.

**Informed Consent Statement:** Not applicable.

**Data Availability Statement:** Data supporting reported results can be found at https://teses.usp.br/teses/disponiveis/16/16135/tde-09062017-110211/pt-br.php (accessed on 15 August 2023).

**Acknowledgments:** The authors gratefully acknowledge the support of the Alexander von Humboldt Foundation and the São Paulo Research Foundation.

**Conflicts of Interest:** The authors declare no conflict of interest.

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
