# Peer review of "Social Landscape, Peripheral Inclusion and Un-Practice: Concepts for Understanding Social Housing Daily Life in Open Spaces"

_sustainability, doi:10.3390/su151712672_

Round 1
Reviewer 1 Report
This manuscript studies the link between urban open spaces and social policies in vulnerable neighbourhoods in Latin America, with an emphasis on their importance for the communities that make them up.
The manuscript addresses an issue of great social interest, however, there are some questions that are not sufficiently answered and some aspects that need to be complemented:
- The title of the manuscript is long
- In the abstract, the objective of the article should be clear and the methodology used should be minimally described.
- The term "Global South" should be written with the first letter in capital letters.
- At the end of the Introduction, the three concepts that are the subject of this manuscript are detailed, however, it is also necessary to write a clear and concise objective that relates these three concepts.
- The methodology described is of little scientific interest, so it is necessary to rely on studies that allow objective data to be analysed.
- The article could be improved if the case studies in Latin America were compared with analogous neighbourhoods elsewhere. For example, the Barrio de San Pablo (Seville, Spain), a social housing condominium built during the Spanish dictatorship to accommodate rural-urban migration:
Núñez-Camarena, G.M.; Clavijo-Núñez, S.; Rey-Pérez, J.; Aladro-Prieto, J.-M.; Roa-Fernández, J. Memory and Identity: Citizen Perception in the Processes of Heritage Enhancement and Regeneration in Obsolete Neighborhoods—The Case of Polígono de San Pablo, Seville. Land 2023, 12, 1234. https://doi.org/10.3390/land12061234
- Establishing a comparison with the criteria analysed may make this study more interesting.
- The manuscript should have a final section of Conclusions.
This manuscript is very interesting and deals with a subject of great social interest; however, it is necessary to provide it with more scientific instrumentation, for example, by making a comparison with cases of similar studies.
Minor editing of English language required
Author Response
Dear reviewer of the Manuscript ID sustainability-2530887,
First of all, we would like to thank you for reviewing our manuscript. We appreciate the received comments, which we considered to improve our paper.
In respect to your time and effort, we are sending the reviewed article with the changes in red, to facilitate a new revision process. We also added information as suggested from you and the other reviewers.
Considering your comments on our paper, we would like to highlight:
1- The title of the manuscript was reduced and the abstract was rewritten, also adding the methodology as recommended;
2- We made some changes in the way the research questions are presented and added more information about the method. International case studies and comparisons have also been added;
3- In order to apply the concepts and make the article clearer for readers, we decided to add an empirical part with two case studies. We believe this addition has greatly improved the article;
4- Some spelling and grammar corrections have been made;
5- At the end of the Introduction, where the three concepts are described in detail, additional paragraphs have been added to relate the concepts to each other. The conclusion was also rewritten to return to the main concepts and themes addressed in the introduction.
We would again like to thank you for your contribution in this manuscript. We remain at your disposal if further revisions are required.
Cordially,
The authors.
Reviewer 2 Report
The article under review raises the interesting issue of conceptual analysis to understand social relations in sensitive urban spaces and the importance of including open spaces in social housing policy, as well as the issue of vulnerability and inequalities in cities.
The article clearly defines the research problem and presents the research gap that this research fills. It has been shown that for the countries of the South, due to their specificity, it is necessary to create one's own original concepts describing their reality. One of the topics discussed in the publication is the analysis of the impact of space on the quality of life in social housing.
The article is qualitative, more intuitive and is based on a juxtaposition of theories relating to aspects of living in social housing with the experience gained by the author/authors during doctoral research in social housing in São Paulo-SP and Santiago in 2013-2017. Thanks to this, the reviewed publication is a very good source of information on life and its quality in social housing, and an attempt to indicate the conditions that affect it the most.
The above advantage of the work is also its certain drawback. The article is based on the authors' observations, which may be subjective. The article lacks an empirical (quantitative) part, which could be done by, for example, a simple analysis of surveys among the inhabitants of these areas.
The article also includes a few minor workshop errors that need to be corrected (see: lines 315-316).
Despite some reservations, the whole should be considered an extremely interesting study of the quality and conditions of life in the cities of the so-called poor South. It is ready for publication after minor corrections. In the future, I suggest the authors conduct surveys that would further confirm the conclusions of this research.
Author Response
Dear reviewer of the Manuscript ID sustainability-2530887,
First of all, we would like to thank you for reviewing our manuscript. We appreciate the received comments, which we considered to improve our paper.
In respect to your time and effort, we are sending the reviewed article with the changes in red, to facilitate a new revision process. We also added information as suggested from you and the other reviewers.
Considering your comments on our paper, we would like to emphasize that in order to apply the concepts and make the article clearer for readers, we decided to add an empirical part with two case studies. We believe that this addition has greatly improved the article.
We also made some changes in the way the research questions are presented and added more information about the methodology. International case studies and comparisons have also been added.
At the end of the introduction, where the three concepts are described in detail, additional paragraphs have been added to relate the concepts to each other. The conclusion has also been rewritten to return to the main concepts and themes addressed in the introduction.
Some spelling and grammar corrections have also been made.
We would again like to thank you for your contribution in this manuscript. We remain at your disposal if further revisions are required.
Cordially,
The authors.
Reviewer 3 Report
The authors state that the result of their studies come from examination of social housing in Brazil and Chile but nothing specific is said about those. In what are they so specific and so special respect to other peripheries or "banlieues" or social/racial "ghettos" around the world?
I have the feeling that the authors are mainly quoting their bibliographic sources without adding much data and/or original contribution to the topic. A Ph.D. dissertation cannot be simply turned into a scientific paper -submitted to a journal- without further personal investigation that produces new and original data.
Also the concept of "condominium" might not be the same in every country, so a local definition (and explanation) is necessary.
Author Response
Dear reviewer of the Manuscript ID sustainability-2530887,
First of all, we would like to thank you for reviewing our manuscript. We appreciate the received comments, which we considered to improve our paper.
In respect to your time and effort, we are sending the reviewed article with the changes in red, to facilitate a new revision process. We also added information as suggested from you and the other reviewers.
Considering your comments on our paper, we would like to emphasize that we have added additional explanations on the specificity of the Brazilian and Chilean housing policies and related them to other case studies from around the world. The concept of "condominium" has also been explained and related with the Latin American reality.
Additionally, in order to apply the concepts and make the article clearer for readers, we decided to add an empirical part with two case studies. We believe that this addition has greatly improved the article.
We also made some changes in the way the research questions are presented and added more information about the methodology. International case studies and comparisons have also been added.
At the end of the introduction, where the three concepts are described in detail, additional paragraphs have been added to relate the concepts to each other. The conclusion has also been rewritten to return to the main concepts and themes addressed in the introduction.
Some spelling and grammar corrections have also been made.
We would again like to thank you for your contribution in this manuscript. We remain at your disposal if further revisions are required.
Cordially,
The authors.
Round 2
Reviewer 1 Report
The modifications introduced by the authors have substantially improved the article.
Figure 2 should be improved.